# DStruct2Design: Data Structure Driven Generative Floor Plan Design

## Abstract

Text conditioned generative models for images have yielded impressive results. Text conditioned floorplan generation as a special type of raster image generation task has also received particular attention. However there are many use cases in floorplan generation where numerical properties of the generated result are more important than the aesthetics. For instance, one might want to specify sizes for certain rooms in a floorplan and compare the generated floorplan with given specifications. Current approaches, datasets and commonly used evaluations do not support these kinds of constraints. As such, an attractive strategy is to use as input a data structure which directly encodes those constraints and can therefore be conditioned on to generate the final floorplan. To explore this setting we (1) convert popular image based floorplan datasets into text format for this data-structure to data-structure formulation of floorplan generation using two popular image based floorplan datasets RPLAN and ProcTHOR-10k, and provide the tools to convert further procedurally generated ProcTHOR floorplan data into our format. (2) We explore the task of floorplan generation given a partial or complete set of constraints and we design a series of metrics and benchmarks to enable evaluating how well samples generated from models respect the constraints. (3) We create multiple baselines by finetuning a large language model (LLM), Llama3, and demonstrate the feasibility of using floorplan data structure conditioned LLMs for the problem of floorplan generation respecting numerical constraints. We hope that our language-based approach to this image-based design problem and our newly developed benchmarks will encourage further research on different ways to improve the performance of LLMs and other generative modelling techniques for generating designs where quantitative constraints are only partially specified, but must be respected.

## 1 Introduction

Generative modelling has the potential to accelerate and improve design tasks but in order for them to achieve widespread use by real-world practitioners they need to be flexible and exhibit a sufficiently high degree of controllability. For instance, in floorplan generation, users will want to be able to specify a set of constraints detailing how rooms should be connected and what their dimensions are. The resulting model should also be flexible in the sense that it performs equally well on as few or as many constraints as needed. Just as there are many possible constraints that the user can specify, there also needs to be a rigorously-defined set of evaluation metrics to probe how well the model performs (or underperforms) on them, which we contribute in this work.

Most existing approaches to floorplan-based generative models focus on 2D top-down renders. These methods typically specify room connectivity via bubble diagrams as well as other information such as room geometries. However, these methods fall short in various areas, for instance some produce raster-based renderings of the floorplan (e.g. vision-based generative models) which make it difficult to directly access metadata. Others may only support the specification of adjacencies instead of also supporting room geometry, limiting the controllability of the model. In this work, we instead opt to represent generated floorplans as a JSON-based data structure specifying all the required metadata of the floorplan as well as specifying each room as a list of polygons (similar to Shabani et al. (2022)). These floorplans are a result of consolidating two existing

large floorplan datasets, namely RPLAN and ProcTHOR. Furthermore, we validate the use of such a dataset by training an LLM to produce such a structure conditioned on constraints specified in natural language. Therefore, our contributions are as follows:

1. We contribute a unified floorplan dataset by merging existing popular datasets (RPLAN and ProcTHOR) and postprocessing them into a format which is amenable to language-based generative models which take as input and output structured text.

2. We design an array of metrics to benchmark the performance of the generated floorplans, in particular how well they respect the constraints set by the user.

3. We train and evaluate an LLM on our task setting to demonstrate the feasibility of our proposed problem formulation.

## 2 Related Work

**Datasets.** One of the most commonly used datasets for floorplan generation is RPLAN (Wu et al., 2019). Since we make use of the dataset in this work, we defer its description to Section 3.2. LIFULL (National Institute of Informatics, 2015) is an extremely large-scale dataset consisting of floorplans sourced from a Japanese realtor company. Floorplans are provided in raster format, however a subset of vectorized versions exist (Liu et al., 2017). One common shortcoming is that there is a lack of well-annotated floorplan data (e.g. vectorised) which is amenable to training. Architext (Galanos et al., 2023) address this issue by generating synthetic but diverse floorplans via a CAD script for Rhinoceros 3D. In a similar vein, we leverage ProcTHOR (Deitke et al., 2022) (also used and described in more detail in Section 3.2) which is a fully open source simulator of buildings (and by extension floorplans). Unlike Rhinoceros 3D this simulator is completely open source.

**Generative Models.** *House-GAN* and *House-GAN++* (Nauata et al., 2020a; 2021a) are a family of GAN-based methods which learn to generate floorplans using convolutional graph-based networks. Notably however this line of work only conditions on a bubble graph. *FloorplanGAN* (Luo and Huang, 2022) proposes a self-attention-based GAN which takes as input (for each room) room centers, desired areas (encoded as relative proportions), as well as room type for each room. However, output from the GAM do not always respect the original constraints. While the use of a differentiable rasteriser to compute losses in pixel space opens up possibilities, the core method does not appear to support a partial specification of constraints or polygons. *HouseDiffusion* (Shabani et al., 2023) is a diffusion-based method which directly predicts a list of polygons for each room, utilising a transformer architecture which also conditions on a bubble diagram. [1]Lastly, *ArchiText* (Galanos et al., 2023) also leverages LLMs to generate output floorplans but it appears the prompts are only limited to natural language descriptions rather than geometry.

**3D Scene Generation.** Recently, full 3D scene generation methods have shown impressive results. AnyHome (Wen et al., 2023) and Holodeck (Yang et al., 2024) are able to generate floor plans, windows, doors, furniture and meaningful placement in 3D all from a natural language query (e.g. 'a 1b1b apartment'). In our method, we focus only on the floor plan aspect, but we allow for specification of room dimensions and areas as well as total floor plan area, which neither method does. Since our proposed dataset is partially comprised of ProcTHOR (a 3D-based floorplan simulator), in principle a future version of our proposed dataset could support the placement of furniture and other props. An overview of how our method compares to the others can be found in Tab. 1.

## 3 Our Text-Based DStruct2Design Floorplan Dataset

In this work, we view the problem of floor plan generation from a text-based data structure perspective. Specifically, we wish to utilize the data structure to enable, through language input, a unified method that

---

[1]While this method could in principle be conditioned on an initial polygonal room specification (via partial noising and subsequent denoising), from the point of view of pure inference (that is, starting from pure noise) *HouseDiffusion*'s only form of conditioning is via a bubble diagram.

| Method | Input | Output | Can specify Geometry? | Absolute Coordinates? |
|---|---|---|---|---|
| House-GAN, House-GAN++, HouseDiffusion | Bubble diagram (room adjacency) | Room masks | no | no |
| ArchiText | Natural Language | Room polygon coords | no | yes |
| FloorplanGAN | Fixed-Length Tensor | Room masks | yes | no |
| AnyHome, Holodeck | Natural Language | Furnished 3D models | no | yes |
| **Ours** | **Structured Language** | **Room polygon coords** | **yes** | **yes** |

Table 1: **Comparison to related works.** Our method focuses on generation of floor plans as opposed to full 3D scenes, and it allows the user to specify individual room geometry (height, width, and/or area) and global apartment geometry in meters and sq. meters.

not only allows users to apply numerical constraints but also retains the ability to condition on graphs such as bubble diagrams, which have been the input in the traditional floorplan generation task. As such, the choice of data representation for the floorplans is very important. In our work, we carefully design a JSON based data structure to be used as a new representation for the floorplans. As prior work all use datasets of floorplans in 2D image or 3D scenes, we create a new dataset by converting existing image and scene data into our data structure for the use of our task.

### 3.1  Data Structure

In this structure, we define crucial numerical data such as the number of rooms present, the total area and type of each room that appears in each floor plan. Each room within a floor plan is further defined with room specific fields. Importantly, each room's location is defined through a set of vertices that forms a polygon. This ensures that the language model cannot cheat by outputting vague locations, and instead it has to predict the exact coordinates of the vertices. The specific structure is presented in Table 2, and a full example of floorplan in this structure is presented in the Appendix J.

This structure also has several additional advantages: **1.** Using numerical values gives us the ability to set numerical constraints in the input, and it allows for clear and explicit evaluation on how well the generation process adheres to these constraints. **2.** This structure is designed to have sufficient information to be transformed into higher level representations such as floor plan images, allowing seamless integration with traditional visualization techniques. **3.** The format is highly extensible, enabling the inclusion of additional information in the generation process, such as the placement and attributes of objects within the floorplan. This flexibility ensures that our approach can be further adapted to various task expansions in the future.

### 3.2  Datasets & Preparation for Our New Task Formulation

**ProcTHOR-10k** (Deitke et al., 2022) is a dataset of 12,000 procedural generated, fully interactive 3D houses designed for research in Embodied AI. We clean and process the raw data from each house, focusing on the geometric properties of the rooms. We employ the shoelace formula to calculate areas and determined room dimensions based on their $x$ and $y$ coordinates. Additionally, we remove redundant points and apply rounding to the coordinates for consistency. Next, we categorize and count the types of rooms, and compute the total area for each house layout. The processed data is organized into the new JSON structure explained in Table 2.

**RPLAN** (Wu et al., 2019) is a manually collected dataset of 80,788 real world floor plans of buildings in Asia. Each floor plan in RPLAN is stored as a $256 \times 256 \times 4$ vector image. Channels 1 and 2 store interior and exterior boundary information; channel 3 contains room information where each pixel value denotes which room it belongs to; channel 4 has extra information to distinguish rooms with the same room type

| | |
|---|---|
| **"room_count"** | ▷ number of rooms |
| **"total_area"** | ▷ total area in square units |
| **"room_types"** [ ] | ▷ list of string of room types present |
| **"rooms"** [ ] | ▷ list of individual room dictionary containing room specifics |
| **"area"** | ▷ area of this room |
| **"floor_polygon"** [ ] | ▷ list of vertices that defines this room's layout |
| **"x"** | ▷ x coordinate of the vertex |
| **"y"** | ▷ y coordinate of the vertex |
| **"is_regular"** | ▷ flag that indicates if the room's shape is rectangular |
| **"height"** | ▷ the y-axis length of the rectangle bounding box enclosing the room |
| **"width"** | ▷ the x-axis length of the rectangle bounding box enclosing the room |
| **"id"** | ▷ unique id of this room |
| **"room_type"** | ▷ the type of this room |
| **"edges"** [ ] | ▷ defines the connection between rooms for bubble gram generation |

Table 2: Processed Floorplan Structure

value in channel 3. To convert this 4 channel image into a JSON structure with well-defined room location, we first extract all pixel coordinates for all of the rooms. For each room, we locate the pixels that make up its perimeter. Then, by tracing the perimeter in one direction we are able to capture all of the vertices in an order that allows recreation of the room polygon. We take the room's type and deduce all the other fields listed in Table 2 to complete the data structure. We convert 80,315 floorplans from RPLAN.

## 3.3 Bubble Diagrams

Traditionally, much of the prior work has assumed that at the beginning of the floor plan design process a bubble diagram is used to conceptualize the layout of the floor plan. As shown in Figure 2, a bubble diagram is used to represent different room and their spacial relationship with one another as a guidance and constraint for the floor plan design process to follow. Formally, a bubble diagram is a graph $\mathcal{G} = (N, E)$ where each node $n_k$ in $N = \{n_j\}_1^R$ represents the $k^{th}$ room in the floor plan with $R$ rooms, and each edge $e_i = (n_p, n_q)$ in $E = \{e_j\}_1^M$ denotes a connection between room $p$ and $q$.

In this work, we enable the use of bubble diagrams as an additional conditioning in our floor plan generation process. We obtain the bubble diagram using pair-wise proximity rooms in each floor plan. If the Manhattan distance between the rooms' boundaries are within a threshold, they are counted as connected. This threshold is set to 8 pixels for the RPLAN dataset in line with prior work, and to 2 pixels for ProcTHOR to account for the length unit differences between the two datasets. This adjacency information is stored in the `edges` field in our data representation. During training, we use $\mathcal{G}$ as conditioning for our floor plan generation.

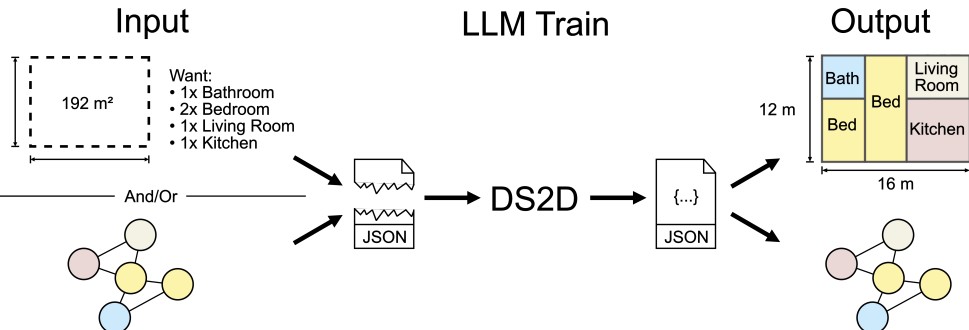

Figure 1: Model Pipeline Overview

# 4 Our DStruct2Design (DS2D) LLM

We train an LLM by prompting it with structured numerical constraints and or graph constraints (Bubble Diagram) and asking it to predict the converted floor plan, as depicted in Figure 1.

**Numerical Constraints** $\mathcal{C} = \{c_j\}_1^T$ are a set of $T$ conditions that users may impose on the final floor plan. In the task of floor plan design and generation, these can include, but are not limited to, total square footage of the entire floor, the number and the types of rooms present, the size of each room. These conditions are mostly inherently numerical, and thus we can take advantage by directly using it in its data structure form. For example, the constraint of having a total square footage of 990 can be transformed into `{"total_area": 990}`. This allows a direct match between the input prompt and output JSON string, which may help the model understand the structure and its relationship better.

**Bubble Diagrams** are represented differently as conditioning to our model. We chose to pass it in as tuples of connecting rooms. For an edge $e_i = (n_p, n_q)$ connecting nodes $p$ and $q$, it is formatted as (`room_p, room_q`). Due to the possible existence of multiple rooms of the same type, which can be clearly distinguished in a graph but not easily in text, we decide to represent `room_p` and `room_q` by their room IDs in addition to their room types.

We process the constraint set and the bubble diagram into "constraint string" and "adjacency string". They are then combined with an instruction phrase that is used throughout the training process to create the final prompt. An example of the full prompt is shown in Appendix C

# 5 Our Metrics and Benchmarks

To assess floorplan quality, we focus on the numerical consistency of the generation. Specifically, we design two groups of metrics to evaluate what we call self-consistency and prompt-consistency:

- **Self Consistency** metrics measures how numbers agree with each other in the generated floor plan. For instance, this includes a metric to check if the area defined by the polygon vertices is the same as the area number presented in the `room_area` field.

- **Prompt Consistency** metrics evaluate how consistent the generation is to the constraints used in the prompt. An example of this type of metrics is one that measures if the generated number of rooms adds up to the number of room requested in the prompt.

In addition to the two main groups of metrics, we also incorporate **Compatibility** metrics which is used in past work (Shabani et al., 2022; Nauata et al., 2020b; 2021b; Johnson et al., 2018; Ashual and Wolf, 2019) to measure similarity between the input bubble diagram and the output floor plan. It is by definition the graph edit distance (Abu-Aisheh et al., 2015) between the input bubble diagram and the output diagram extracted from the output JSON. The extraction method is the same one used to generate the bubble diagrams in the first place as described in Section 3.3. The full array of metrics and their explanations are presented in Table 3.

# 6 Experiments

All of our experiments are ran on the LLaMA3-8B-Instruct variant of the LLaMA model family (Touvron et al., 2023). We train our models by running 8-bit quantization along with LoRA (Hu et al., 2021). Each of our experiments are also run on single RTX8000 GPU, they take less than a day to complete. The exact training parameters and data split details can be found in Appendix D

## 6.1 Model Variants

We train six variants of our model on the converted ProcTHOR dataset and four on converted RPLAN dataset. Each of the four models on RPLAN – **5-R**, **6-R**, **7-R**, **8-R** – is trained without floorplan data of a

| Metric Type | Metric | Explanation |
|---|---|---|
| Self Consistency (SC) | † Total Area | percentage difference between stated total area vs. the sum of all rooms' state area |
| | ‡ Polygon Area (P. Area) | percentage difference between a room's stated area vs. area calculated from polygon |
| | ♯ Overlap | boolean check for existence of polygon overlap |
| | ♭ Room ID (ID) | boolean check for duplicate room id |
| | ♮ Room Count (R. Count) | boolean check to see if "room_count" field number is equal to the number of rooms in the floor plan. |
| Both SC and PC | ◁ Room Height (R. H) | percentage difference between height obtained from polygon vs. stated height in generation (SC) or requested height (PC). |
| | ▷ Room Width (R. W) | percentage difference between width obtained from polygon vs. stated width in generation (SC) or requested width (PC). |
| Prompt Consistency (PC) | ● Total Area | percentage difference between the sum of room polygon areas vs. requested total area |
| | ♣ Num. Room (Num. R.) | percentage difference between number of rooms in the floorplan vs. requested number of rooms in the prompt. |
| | ♠ Room ID (ID) | precision and recall of room ids in the floorplan with respect to the ids present in the prompt |
| | ♡ Room Area (R. Area) | percentage difference between a room's polygon area vs. requested room area in the prompt |
| | ♢ Room Type (Type) | precision and recall of room types in the floorplan with respect to the room types present in the prompt |
| | ○ ID-Type Match (IDvsType) | percentage difference between the number of rooms with correctly matching type and id in the floor plan vs. the total number of rooms present in the prompt |
| Bubble Graph | Compatibility | graph edit distance (GED) between the input bubble diagram and one extracted from the output floor plan. |

Table 3: Metrics and Associated Acronyms Used to Evaluate Generation Quality

| | | ProcTHOR Models | | | | | RPLAN Models | | | |
|---|---|---|---|---|---|---|---|---|---|---|
| | F | M | PM | F+BD | M+BD | PM+BD | 5-R | 6-R | 7-R | 8-R |
| Bubble Diagram Masking | - | - | - | ✓ | ✓ | ✓ | ✓ | ✓ | ✓ | ✓ |
| | - | random | preset | - | random | preset | random | random | random | random |

Table 4: Summary of model variants

certain room count. 5-R is trained only on floorplans with 6, 7, and 8 rooms, etc. This follows prior work (Nauata et al., 2020b; 2021b; Shabani et al., 2023). On the other hand, the six models trained on ProcTHOR are divided two main variants: bubble diagram enabled model, and numerical constraint only model. In the **bubble diagram enabled model (BD)**, both the constraint string and the adjacency string described in Section 4 are used as part of the input. In constrast, in the **numerical constraint only model**, only the constraint string is used. We further train three sub-variants for each of the two main variants to investigate how robust the models are to missing information:

- **Full-Prompt (F)** model takes advantage of every attribute available in the constraint set. During training, every constraint attribute is part of the constraint string.

- **Mask (M)** model uses a subset of constraint set by applying a 50% masking on every single possible constraint. In the case of `rooms` which is an attribute containing a list, the masking is applied independently to individual item in the list. (`rooms` is only dropped out if the entire list becomes empty.) As a safety measure, we always keep at least 1 constraint in the set.

- **Preset Mask (PM)** randomly selects one of four preset constraint sets with varying degree of missing information. The idea is to not just have IID random masking, but a hierarchy of attributes from general to specific which are detailed in Table 11 in Appendix F.

The summary of our six variants and their abbreviations are shown in Table 4

## 6.2 Generation Prompts

To test the model, we run four different generation prompts with varying amount of constraints in the prompt. This simulates the real world problem of floorplan design, where the designer often receives only partial criteria. For instance, the user might only ask the living room and bedroom to be a certain size, and leaves the designer with the freedom to imagine layouts with varying sizes for the other rooms. The four generation prompts are listed in order of decreasing amount of constraints:

- **Specific (S)** is one where we use all the possible constraints

- **All Room Area (AR)** is one where we pass in the area of all the rooms (total area can be inferred from this information).

- **Partial Room Area (PR)** is similar to Total Area. In addition to the total area of the floor plan, the area of some of the rooms are also passed as conditions in the constraint string.

- **Total Area (TA)** is one where we only use the total area of the floor plan as constraint.

For all of the Bubble Diagram model variants, Bubble Diagrams are used in each of these types of prompts.

## 7  Results

We test floor plan generation quality on ProcTHOR-trained six model variants with the four different generation prompts with self and prompt consistency metrics. Results from our best model variants are listed in Table 5, and all results on all 48 sets of experiments are listed in Appendix G. We also perform the same evaluation on the four RPLAN-trained model variants. Because each model variant trained on RPLAN uses a slightly different set of data (following prior work), we show results for each model variant in Table 5. Results for the full 24 experiments are shown in Appendix I.

**Results Analysis**   Our LLM models demonstrate a high level of competence in generating floorplans that are self consistent, and are also consistent with input numerical constraint. As seen in Figure 2, when given full specifications, the generated rooms largely resembles ground truth in sizes, width and height. This quality is accurately reflected by our Total Area, R.H and R.W metrics. On the other hand, our metric suggests that generations can produce overlapping room layouts. An example of this is found in Figure 2, in the second row's green background's right-most floorplan: the top-left bathroom is overlapping on top of the bedroom. This problem is especially apparent in the RPLAN-trained models, and is accurately reflected by the P. Overlap metrics in Table 5. This behaviour is somewhat expected as LLMs are known to struggle in understanding geometry Gao et al. (2023); Zhang et al. (2024). This may perhaps be tackled with chain of thought reasoning in future work. However our LLM approach is able to produce floorplan with numerical values consistent within itself and with respect to the input constraint, and it shows that an LLM approach to solving the floorplan design problem is indeed viable.

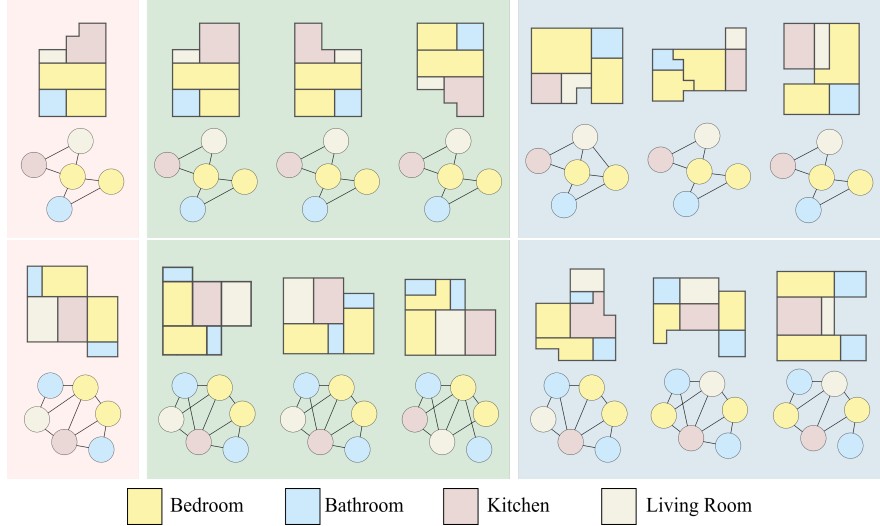

Bedroom    Bathroom    Kitchen    Living Room

Figure 2: Pink background: ground truth floor plan and ground truth bubble diagram. Green background: generated floor plan using Specific prompts, and extracted bubble diagram. Blue background: generated floor plan using Total Area prompts, and extracted bubble diagrams.

| | **Self-consistency Benchmark** | | | | | | |
|---|---|---|---|---|---|---|---|
| | P. Area$^\ddagger$ | Total Area$^\dagger$ | P. Overlap$^\sharp$ | R. Count$^\sharp$ | ID$^\flat$ | R. H$^\triangleleft$ | R. W$^\triangleright$ |
| $F_S$ | 0.94±0.07 | 1.00±0.03 | 0.16±0.37 | 1.00 | 1.00 | 0.99±0.02 | 0.99±0.02 |
| $M_{AR}$ | 0.90±0.07 | 0.99±0.03 | 0.12±0.33 | 1.00 | 1.00 | 0.99±0.02 | 0.99±0.01 |
| $PM_{PR}$ | 0.92±0.06 | 0.97±0.07 | 0.15±0.35 | 0.96 | 1.00 | 1.00±0.01 | 1.00±0.01 |
| $PM_{TA}$ | 0.93±0.05 | 0.99±0.04 | 0.14±0.35 | 0.95 | 1.00 | 1.00±0.01 | 1.00±0.01 |
| 5-$R_{AR}$ | 0.90±0.16 | 0.98±0.05 | 0.37±0.48 | 0.97 | 1.00 | 1.00±0.01 | 0.98±0.02 |
| 6-$R_S$ | 0.93±0.05 | 0.98±0.06 | 0.36±0.48 | 0.99 | 1.00 | 0.99±0.01 | 0.97±0.02 |
| 7-$R_S$ | 0.93±0.04 | 0.98±0.05 | 0.42±0.49 | 0.97 | 1.00 | 0.99±0.01 | 0.98±0.02 |
| 8-$R_{TA}$ | 0.82±3.50 | 0.97±0.07 | 0.53±0.50 | 0.89 | 1.00 | 0.99±0.01 | 0.98±0.02 |
| | **Prompt-consistency Benchmark** | | | | | | |
| | Num. R$^\clubsuit$ | Total Area$^\bullet$ | R. Area$^\heartsuit$ | ID$^\spadesuit$ | IDvsType$^\circ$ | R. H$^\triangleleft$ | R. W$^\triangleright$ |
| $F_S$ | 1.00±0.00 | 0.95±0.07 | 0.94±0.07 | 1.00±0.03 | 1.00±0.00 | 0.99±0.02 | 0.99±0.02 |
| $M_{AR}$ | 1.00±0.00 | - | 0.90±0.07 | 1.00±0.00 | 1.00±0.00 | - | - |
| $PM_{PR}$ | 1.00±0.00 | 0.93±0.07 | 0.91±0.11 | 1.00±0.03 | 1.00±0.00 | - | - |
| $PM_{TA}$ | 1.00±0.00 | 0.95±0.06 | - | - | - | - | - |
| 5-$R_{AR}$ | - | - | 0.92±0.04 | 1.00±0.01 | 1.00±0.00 | - | - |
| 6-$R_S$ | 1.00±0.00 | 0.94±0.07 | 0.94±0.06 | 1.00±0.05 | 1.00±0.00 | 0.92±0.23 | 0.94±0.14 |
| 7-$R_S$ | 1.00±0.00 | 0.94±0.06 | 0.94±0.06 | 1.00±0.02 | 1.00±0.00 | 0.93±0.17 | 0.94±0.29 |
| 8-$R_{TA}$ | - | 0.93±0.07 | - | - | - | - | - |

Table 5: Our self and prompt-consistency benchmark results. Subscripts under model variants are generation prompts.

To compare with prior work that focus on using bubble diagrams as input on the RPLAN dataset, we prompt our RPLAN-trained models with bubble diagram. For evaluation we compute the standard compatibility metric obtained from graph edit distance calculation between the input bubble diagrams and the ones extracted from generations. Results are shown in Table 6. We also evaluate compatibility metrics on the ProcTHOR-trained BD model variants. The results are shown in Table 7.

| | Number of Rooms in Floorplan | | | |
|---|---|---|---|---|
| Model | 5 | 6 | 7 | 8 |
| Ashual and Wolf (2019) | 7.5 ± 0.0 | 9.2 ± 0.0 | 10.0 ± 0.0 | 11.8 ± 0.0 |
| Johnson et al. (2018) | 7.7 ± 0.0 | 6.5 ± 0.0 | 10.2 ± 0.0 | 11.3 ± 0.1 |
| House-GAN (Nauata et al., 2020a) | 2.5 ± 0.1 | 2.4 ± 0.1 | 3.2 ± 0.0 | 5.3 ± 0.0 |
| House-GAN++ (Nauata et al., 2021a) | 1.9 ± 0.3 | 2.2 ± 0.3 | 2.4 ± 0.3 | 3.9 ± 0.5 |
| HouseDiffusion (Shabani et al., 2023) | 1.5 ± 0.0 | 1.2 ± 0.0 | 1.7 ± 0.0 | **2.5** ± 0.0 |
| Our BD Model w/ BD prompt | **0.46** ± 0.73 | **0.79** ± 0.98 | **1.27** ± 1.28 | **2.50** ± 1.97 |

Table 6: Comparison of Compatibility↓ score (Graph Edit Distance) on the RPLAN dataset.

**Compatibility Analysis**  Overall, our RPLAN-trained model generates results that are competitive with the state-of-the-art image based algorithms, even though our focus here has been on numerical accuracy. One caveat is that as the complexity of the generations increases (for example, the number of rooms increases, and the generations becomes much longer), the fine-tuned model starts to create some extreme floorplans. It would generate floorplans with a lot more overlaps causing the bubble diagram to be very different than the one used as constraints. In this sense, when there are more rooms in a floorplan, overlapping becomes a bigger issue, and its effect on the bubble diagram becomes more apparent. This causes our method to have a much higher variance in the compatibility metrics compared to existing image-based approaches. Table 7 suggests our model has a harder time following the input bubble diagram as the number of room increases, or when given room area information. This is reflected in Figure 2 where the second row's bubble diagrams show more variations. This limitation might be tackled by using a better data structure to represent bubble diagrams, and perhaps also to be included as part of the floorplan data representation. This way, the bubble diagram will simply be part of the constraint set, and this may help the model understand the input better. This can be explored in future work in the same task setting. Additionally, in the future, our data can be expanded to account for more floorplan attributes such as doors and walls and our benchmark can include additional metrics to account for more obscure numerical checks.

| Model | Generation | Number of Rooms | | | | |
|---|---|---|---|---|---|---|
| Variant | Prompt | 3 | 4 | 5 | 6 | 7 |
| F+BD | Specific | 0.15 ±0.41 | 0.33 ±0.58 | 1.35 ±1.34 | 2.24 ±1.76 | 4.23 ±1.89 |
| | All Room Area | 5.91 ±0.29 | 9.75 ±0.54 | 13.79 ±1.15 | 18.09 ±1.71 | 22.35 ±2.24 |
| | Partial Room Area | 3.80 ±1.20 | 5.73 ±2.03 | 8.19 ±2.72 | 10.15 ±2.88 | 11.42 ±4.52 |
| | Total Area | 0.19 ±0.40 | 0.62 ±0.78 | 3.48 ±1.83 | 3.41 ±1.50 | 6.23 ±2.20 |
| M+BD | Specific | 0.16 ±0.45 | 0.47 ±0.67 | 2.06 ±1.33 | 2.85 ±1.79 | 5.23 ±2.14 |
| | All Room Area | 5.91 ±0.29 | 9.62 ±0.56 | 12.90 ±1.27 | 17.48 ±1.82 | 22.13 ±1.94 |
| | Partial Room Area | 3.85 ±1.11 | 6.35 ±1.95 | 8.44 ±2.48 | 11.56 ±3.42 | 14.13 ±3.90 |
| | Total Area | 0.20 ±0.40 | 0.54 ±0.73 | 2.40 ±1.03 | 3.09 ±2.07 | 6.26 ±2.03 |
| PM+BD | Specific | 0.15 ±0.39 | 0.58 ±0.96 | 2.42 ±1.33 | 2.97 ±1.62 | 5.97 ±1.25 |
| | All Room Area | 5.88 ±0.33 | 9.77 ±0.49 | 13.17 ±1.23 | 18.56 ±1.58 | 21.97 ±2.36 |
| | Partial Room Area | 3.87 ±1.13 | 6.42 ±1.97 | 8.99 ±2.67 | 12.06 ±3.05 | 13.68 ±3.71 |
| | Total Area | 0.34 ±0.93 | 0.46 ±0.84 | 2.08 ±1.29 | 2.65 ±1.87 | 5.84 ±1.61 |

Table 7: Compatibility↓ score on ProcTHOR-trained models.

**Natural Language Prompt Analysis**  We test our model with natural language prompts where the constraints are given in natural language such as "the floor plan should have 120 squared meters in total area." and "the floor plan should have 3 rooms. It has 1 kitchen, 1 bedroom, 1 bathroom." The results are presented in Table 8. The model trained with natural language prompt achieves high accuracy on metrics associated with general aspects of the floorplan such as total area, and total number of rooms. However, it is more prone to creating overlaps and consistently scores lower in specific room constraints such as height and width.

| | Self-consistency Benchmark | | | | | | |
| --- | --- | --- | --- | --- | --- | --- | --- |
| | P. Area$^\ddagger$ | Total Area$^\dagger$ | P. Overlap$^\sharp$ | R. Count$^\natural$ | ID$^\flat$ | R. H$^\triangleleft$ | R. W$^\triangleright$ |
| NL$_S$ | 0.99±0.00 | 1.00±0.00 | 0.25±0.43 | 1.00 | 1.00 | 0.93±0.00 | 0.93±0.00 |
| NL$_{AR}$ | 1.00±0.00 | 1.00±0.01 | 0.24±0.43 | 1.00 | 1.00 | 0.93±0.00 | 0.93±0.00 |
| NL$_{PR}$ | 1.00±0.00 | 1.00±0.00 | 0.23±0.42 | 0.70 | 1.00 | 0.93±0.00 | 0.93±0.00 |
| NL$_{TA}$ | 1.00±0.00 | 1.00±0.00 | 0.23±0.42 | 1.00 | 1.00 | 0.93±0.00 | 0.93±0.00 |
| | Prompt-consistency Benchmark | | | | | | |
| | Num. R$^\clubsuit$ | Total Area$^\bullet$ | R. Area$^\heartsuit$ | ID$^\spadesuit$ | IDvsType$^\circ$ | R. H$^\triangleleft$ | R. W$^\triangleright$ |
| NL$_S$ | 1.00±0.00 | 1.00±0.00 | 0.99±0.00 | 0.82±0.39 | 1.00±0.00 | 0.93±0.02 | 0.93±0.01 |
| NL$_{AR}$ | 1.00±0.00 | - | 1.00±0.00 | 0.93±0.01 | 0.93±0.01 | - | - |
| NL$_{PR}$ | 1.00±0.00 | 1.00±0.00 | 1.00±0.00 | 0.98±0.14 | 1.00±0.00 | - | - |
| NL$_{TA}$ | 1.00±0.00 | 1.00±0.00 | - | - | - | - | - |

Table 8: Our self- and prompt-consistency on natural language prompt trained model

| | Self-consistency Benchmark | | | | | | |
| --- | --- | --- | --- | --- | --- | --- | --- |
| | P. Area$^\ddagger$ | Total Area$^\dagger$ | P. Overlap$^\sharp$ | R. Count$^\natural$ | ID$^\flat$ | R. H$^\triangleleft$ | R. W$^\triangleright$ |
| Few-Shot$_S$ | 0.99±0.00 | 1.00±0.00 | 0.04±0.20 | 1.00 | 1.00 | 0.93±0.02 | 0.93±0.01 |
| Few-Shot$_{AR}$ | 0.99±0.01 | 0.96±0.07 | 0.01±0.11 | 1.00 | 1.00 | 0.88±0.07 | 0.93±0.00 |
| Few-Shot$_{PR}$ | 1.00±0.00 | 0.94±0.12 | 0.00±0.04 | 0.70 | 1.00 | 0.86±0.07 | 0.93±0.00 |
| Few-Shot$_{TA}$ | 1.00±0.00 | 1.00±0.00 | 0.00±0.00 | 1.00 | 1.00 | 0.93±0.00 | 0.93±0.00 |
| | Prompt-consistency Benchmark | | | | | | |
| | Num. R$^\clubsuit$ | Total Area$^\bullet$ | R. Area$^\heartsuit$ | ID$^\spadesuit$ | IDvsType$^\circ$ | R. H$^\triangleleft$ | R. W$^\triangleright$ |
| Few-Shot$_S$ | 0.82±0.39 | 1.00±0.00 | 0.99±0.00 | 0.82±0.39 | 1.00±0.00 | 0.93±0.02 | 0.93±0.01 |
| Few-Shot$_{AR}$ | 0.95±0.22 | - | 0.99±0.01 | 0.95±0.22 | 1.00±0.00 | - | - |
| Few-Shot$_{PR}$ | 0.98±0.14 | 1.00±0.00 | 1.00±0.00 | 0.98±0.14 | 1.00±0.00 | - | - |
| Few-Shot$_{TA}$ | 1.00±0.03 | 1.00±0.00 | - | - | - | - | - |

Table 9: Our self- and prompt-consistency on few-shot results

**Few-Shot Analysis**  We further present few-shot results with GPT4o-mini. This is done by prompting the LLM with multiple example specifications along with output floorplan Jsons. The results are shown in Table 9. The generated floorplan has less overlapping polygons compared to the ones generated by the fine-tuned LLAMA3 models, however, they are less accurate in following the size, height and width specifications.

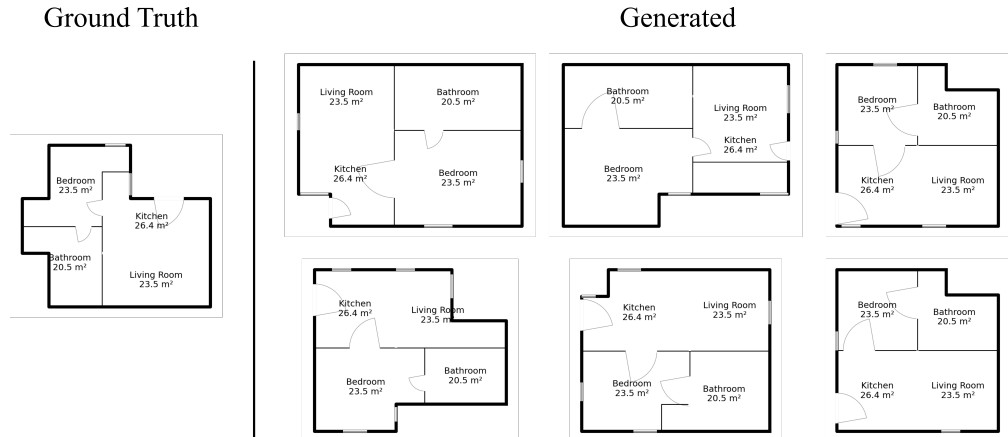

Figure 3: Generated realistic floor plans using a subset of constraints extracted from the ground truth.

| | Self-consistency Benchmark | | | | | | |
|---|---|---|---|---|---|---|---|
| | P. Area$^\ddagger$ | Total Area$^\dagger$ | P. Overlap$^\sharp$ | R. Count$^\natural$ | ID$^\flat$ | R. H$^\triangleleft$ | R. W$^\triangleright$ |
| Window+Doors$_S$ | 0.99±0.01 | 1.00±0.02 | 0.23±0.42 | 1.00 | 1.00 | 0.93±0.00 | 0.93±0.01 |
| Window+Door$_{AR}$ | 0.99±0.00 | 0.99±0.03 | 0.18±0.39 | 1.00 | 1.00 | 0.93±0.00 | 0.93±0.00 |
| Window+Door$_{PR}$ | 0.99±0.00 | 0.98±0.05 | 0.16±0.37 | 0.99 | 1.00 | 0.93±0.00 | 0.93±0.00 |
| Window+Door$_{TA}$ | 0.99±0.00 | 0.99±0.03 | 0.16±0.37 | 1.00 | 1.00 | 0.93±0.00 | 0.93±0.00 |
| | Prompt-consistency Benchmark | | | | | | |
| | Num. R$^\clubsuit$ | Total Area$^\bullet$ | R. Area$^\heartsuit$ | ID$^\spadesuit$ | IDvsType$^\circ$ | R. H$^\triangleleft$ | R. W$^\triangleright$ |
| Window+Doors$_S$ | 1.00±0.00 | 0.99±0.00 | 0.99±0.00 | 1.00±0.02 | 1.00±0.00 | 0.93±0.01 | 0.93±0.01 |
| Window+Door$_{AR}$ | 1.00±0.00 | - | 0.99±0.00 | 1.00±0.02 | 1.00±0.00 | - | - |
| Window+Door$_{PR}$ | 1.00±0.00 | 0.99±0.00 | 0.99±0.00 | 1.00±0.02 | 1.00±0.00 | - | - |
| Window+Door$_{TA}$ | 1.00±0.00 | 0.99±0.00 | - | - | - | - | - |

Table 10: Our windows and doors augmented floorplan results

**Realistic Generation Analysis**   To encourage generation of realistic floorplans, we further add windows, doors, and open concept rooms to our data structure. The evaluation results are shown in Table 10, and some generation reulsts are shown in Figure 3. We can see from the figure that the numerical constraints of the original ground truth floor plan is well respected. Every room as the same total area as the original.

# 8 Conclusions

We have motivated the need for new datasets and benchmarks for real-world use case scenarios for floorplan generation. We have developed and explored a Llama 3 based LLM for the problem and it yields SOTA results on the previous compatibility based evaluation and along with our proposed metrics, it highlights different use cases scenarios where improvements are possible. In particular we see from Table 7 that when conditioning on bubble diagrams and room area information this model struggles to respect the bubble diagram constraints. From Table 5 we see that the model does well, but is far from perfect at generating rooms with correct areas as computed from the generated polygons. We also see that the biggest weakness of this otherwise SOTA model is linked to issues with generating overlapping rooms.

We hope that our data, simulated data generation procedure and benchmarks will stimulate further developments for this new problem formulation for floorplan generation.

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
