# OpenReview forum: "DStruct2Design: Data Structure Driven Generative Floor Plan Design"
_TMLR — Rejected by TMLR_

### Review · Reviewer_Fh9T · 2024-12-10

**Summary Of Contributions:**

This paper introduces an intermediate data structure for LLM-based floor plan generation. The authors designed a JSON-based data structure to fomulate the floor plan, incorporating various properties and constraints. They convert two datasets to this format, then fine-tune an LLM for the floor plan generation task, and design evaluation benchmarks and metrics to validate the effectiveness of their method.

**Audience:**

Yes

**Claims And Evidence:**

No

**Requested Changes:**

Please kindly refer to the weaknesses mentioned above above.

**Strengths And Weaknesses:**

### Strengths

- To the best of the reviewer's knowledge, the application of LLM in Floor Plan design is a novel and impactful advancement. This approach successfully validates the potential of LLMs in spatial reasoning and floor plan design, thereby broadening the scope of LLM applications.
- Experimental results show that the proposed method outperforms baselines across multiple evaluation metrics.

### Weaknesses
1. The manuscript's language is overall informal and unclear, lacking the precision expected in academic papers.  Some example concerns are listed below:

   - "Because the GAN is tasked with also refining the initial input constraints, the output may not respect them." This statement is ambiguous.
   - Use of "mathematically consistent" is vague and undefined.
   - "In this work, we enable the use bubble diagram as an additional conditioning in our floor plan generation process. To obtain the bubble diagram we check pair-wise proximity of the rooms in each floor plan". This sentence contains grammar errors.

   **Requested change**: I suggest the authors revise the manuscript to improve the overall writing of the paper.

2. In the related works section, the authors do not clearly illustrate how relevant floor plan datasets or generative models have developed. Besides, a more thorough literature review of LLMs and Vision Generative Models (GANs, Diffusion Models) should be included.

3. The authors claim to "construct a new dataset"; however, this description is inaccurate as they mainly convert the data format of two existing datasets. Also, in experiments, the authors train their models on the RPLAN and ProcTHOR dataset separately.

   **Requested change**: I suggest the authors revise their terminology. I also suggest the authors to train LLM on jointly both datasets. It would be valuable to investigate whether such joint training enhances performance across both datasets.

4. One of the main contributions mentioned in the paper is the designed JSON-based data structure. While I acknowledge the importance of an effective prompt structure for adapting LLMs to this task, the manuscript lacks experimental results demonstrating the effectiveness of this data structure. E.g. Does this proposed method outperform using natural language input for LLMs?

   **Requested change**: I suggest that the authors include additional ablation studies on different data structures (i.e. the proposed JSON-based data-structure VS LLM taking natural language as input).

5. Lack of clarification on crucial implementation details. How do you convert the bubble diagrams into structured JSON format?

   **Requested change**: Please clarify how to convert input constraints and bubble diagrams to the input json, and how to map the LLM-generated output to the final floor plan or bubble diagram. You may include some examples to enhance the clarity.

6. Lack of qualitative comparisons with other methods.

   **Requested change**: Could you please provide some example use cases of the fine-tuned LLM? Additionally, could provide visual comparison of your method with other baselines (e.g. floor plans generated by House-GAN or HouseDiffusion)?

---

> ### Author Response · Authors · 2025-01-02
> **Response**
>
> We thank the reviewer for their detailed review on different aspects of our work.
>
> **To address requested change 1**:
> - We have made changes to the overall writing of the paper and they will be further refined for the final version
>
> **To address requested change 2**:
> - We will add more details about LLM in the final version
>
> **To address requested change 3**:
> - We are currently training the combined model, and will update the results ASAP.. This was a slower process due to Christmas and New Year being in the middle of our rebuttal period. We would also like to stress that RPLAN, which is a closed source dataset, is included mainly to compare with the existing methods. We believe using PROCThor for its open source, procedurally generated floorplan is a better way for future projects.
>
> **To address requested change 4**:
> - Models using natural language based prompts instead of JSON based prompts are added in Table 8, followed by an analysis and discussion.
>
> **To address requested change 5**:
> - We follow the same bubble diagram conversion method as prior work. We include a detailed explanation in the revision in Appendix E, and will continue to refine it for the final paper.
>
> **To address requested change 6**:
> - We refined Appendix J to include input-output texts of our fine-tuned model, along with image renderings of the model output to aid with understanding the LLM usage. We also include visual comparisons between our generation and those from HouseGAN++ and HouseDiffusion as a qualitative comparison.

---

### Review · Reviewer_Bo6P · 2024-12-18

**Summary Of Contributions:**

* This paper introduced a novel floorplan data conversion method which formats image based floorplan datasets RPLAN and ProcTHOR-10k into a JSON-based structure for precise floorplan representation. This JSON-based structure encoded room geometries, areas, and adjacencies, enabling extensibility and compatibility with numerical and structural constraints.
* The paper developed self-consistency, prompt-consistency, and compatibility metrics to evaluate adherence to constraints and input conditions.
* The paper demonstrated the feasibility of fine-tuned LLaMA3 model for generating constraint-respecting floorplans from structured inputs.
* The paper provided a foundation for language-driven generative floorplan design with benchmarks to guide future improvements.

**Audience:**

Yes

**Broader Impact Concerns:**

The paper already mentioned "Possible Negative Impacts" in Appendix.

**Claims And Evidence:**

Yes

**Requested Changes:**

* Include comprehensive information about the training process, such as the size of the training dataset, number of epochs, batch size, and other relevant hyperparameters, to ensure reproducibility and clarity.
* Conduct a detailed analysis of potential hallucination issues in the LLM's outputs, particularly focusing on scenarios where generated floorplans deviate significantly from specified constraints or exhibit unrealistic features.

**Strengths And Weaknesses:**

Strengths:
* Introduce a novel JSON-based floorplan representation.
* Develop evaluation metrics for self-consistency, prompt-consistency, and compatibility, addressing critical evaluation gaps in floorplan generation.
* Support both numerical and structural constraints, enhancing model usability across diverse design scenarios.

Weaknesses:
* Current evaluations metrics do not fully account for advanced features like object placement or multi-level layouts, which may limit future extensibility.
* The evaluation does not include assessments of the generated floorplans in real-world scenarios or expert judgment, which are critical for validating practical applicability and design quality.
* The paper only fine-tuned LLaMA3 model on the proposed dataset, lacking comparisons with other LLM to mitigate potential model-specific biases and validate generalizability.

---

> ### Author Response · Authors · 2025-01-02
> **Response**
>
> We thank the reviewer for the feedback
>
> **To address the weaknesses**:
> - We understand our current model does not produce objects such as tables and beds and chairs etc. But as a proof of concept, we include placements of windows and doors (Table 9), and we render them out into visual floorplans in Figure 3. Even though our goal is to just create base floorplans, additional objects in a floorplan can be readily extended similar to how windows’ and doors’ placesments are dictated in our generation.
> - Our work primarily focuses on proving the usefulness of using a LLM in this floorplan design problem in order to enable the input of numerical constraints. As such, we mainly focus on using LLAMA3 since it is readily fine-tuneable. We present few-shot learning outputs from GPT4o-mini (Table 8) as a comparison with other LLMs. More detailed comparison can be part of future work that focuses on studying different LLMs’ biases on floorplan generation.
>
> **To address requested change 1**:
> - To ensure reproducibility, we have prepared a github with pretrained model along with instruction on how to run simple commands to train a model from scratch to reproduce the same results. This will be made public after the anonymized review period. Appendix D also outlines more specific training details such as dataset split and hyperparameter choices.
>
> **To address requested change 2**:
> - This issue has been expanded on in the analysis section that addresses a similar concern brought up by reviewer x139 (requested change 2).

---

### Review · Reviewer_x139 · 2024-12-20

**Summary Of Contributions:**

This paper explores the application of large language models (LLMs) for structured data generation, specifically in the domain of floor plan generation. The key contributions include:
1. Proposing a text-based representation for floor plan data and numerical constraints (e.g., total area, room sizes) and graph-based adjacency relationships (bubble diagrams), enabling the generation of floor plans via interpretable JSON-based outputs.
2. Introducing new evaluation metrics to assess the numerical correctness and logical consistency of generated floor plans.
3. Conducting experiments to demonstrate the model’s ability to satisfy both numerical and spatial constraints, showcasing the feasibility of applying LLMs to structured data generation.

**Audience:**

Yes

**Claims And Evidence:**

Yes

**Requested Changes:**

1. Please conduct an ablation study to analyze the impact of different LLM backbones and model scales on the performance of the proposed approach. This will help clarify the method’s dependency on specific LLM characteristics and provide insights into its performance upper bound.
2. Please expand the comparison with referenced models by evaluating the self-consistency metrics on outputs from existing models. This will help demonstrate the effectiveness of the LLM-based approach and validate the informativeness of the self-consistency score. Additionally, please include an in-depth analysis of the trends observed in Table 6, particularly the increase in compatibility scores and uncertainties as task complexity grows.
3. Please provide more discussion on the numerical reasoning challenges faced by LLMs and explore potential improvements, such as explicit reasoning mechanisms like chain-of-thought prompting, to address conflicts between numerical constraints and spatial relationships.

**Strengths And Weaknesses:**

**Strength**
1. The paper explores a novel application of large language models (LLMs) for structured data generation, extending their capabilities to the domain of floor plan design.
2. The proposed textual representation successfully integrates numerical constraints (e.g., total area, room sizes) and spatial adjacency relationships (bubble diagrams), enabling the generation of controllable and interpretable outputs.
3. The work introduces self-consistency and prompt-consistency metrics to evaluate numerical correctness and adherence to user-specified constraints, addressing an aspect—numerical accuracy of generated data—that has been under-studied in this domain.

**Weakness**
1. The major concern is about the Numerical Reasoning Challenges of LLM
    * While the model generates outputs conditioned on numerical constraints, the numerical reasoning process is opaque. LLMs, though capable of solving numerical tasks under specific conditions, often exhibit fragile and inconsistent performance, particularly for tasks requiring precise calculations or logical constraint satisfaction. The absence of explicit reasoning mechanisms, such as chain-of-thought prompting, makes it unclear how the model resolves conflicts between numerical constraints (e.g., room areas, total area) and spatial relationships (e.g., adjacency), reducing the interpretability of the model’s outputs.
    * The paper lacks an ablation study on the backbone LLM, leaving the impact of model architecture, scale, and design unexplored. Since numerical reasoning capabilities are known to improve with model scale, understanding how the choice of LLM influences performance is critical. Without such an analysis, it remains unclear how dependent the method’s success is on specific LLM characteristics.
    * A specific concerning trend is observed in Table 6, where the compatibility score increases as the number of rooms increases, potentially underperforming some competitors and exhibiting larger uncertainties. This highlights a limitation of LLMs in handling numerical tasks as the problem complexity grows, raising concerns about their robustness and reliability for tasks requiring accurate constraint satisfaction at scale.
2. The self-consistency and prompt-consistency metrics yield scores consistently close to 1.0, suggesting a lack of sensitivity to subtle errors or variations. This reduces their effectiveness in distinguishing between model variants or highlighting limitations.

---

> ### Author Response · Authors · 2025-01-02
> **Response**
>
> We thank the reviewer for their insights on the reasoning challenges of LLMs.
>
> **To address requested change 1**:
> - In this work we fine-tune a LLAMA3 model and also run few-shot learning outputs from GPT4o-mini (Table 8). We found that while LLAMA3 is great at satisfying numerical constraints, GPT4o-mini is better at generating floorplans without overlapping rooms. We believe, as a first work to prove LLM’s usefulness in solving the floorplan design problems with numerical constraints, both of these models accomplish that tasks by generating highly accurate numerical values in the floorplan that agrees with the given constraints.
>
> **To address requested change 2**:
> - The existing approaches do not have a way to incorporate numerical constraints at all, and we cannot compare numerical consistency metric with the existing approaches. The only comparable metric from the existing methods is the “compatibility” metric, which is essentially the edit distance between the input bubble diagram and the final bubble diagram of the generated floorplan. And we present a comparison of this compatibility metric between our generations and those from the existing methods in Table 6.
> - As for the trend observed in Table 6, where the floorplan seems to have more variance in edit distance as complexity grows, it is because when the floorplan complexity increases, the generation starts to have more overlapping rooms, which in turn causes the bubble diagram to become more “wrong”. We expanded the analysis section for this phenomenon and will continue to refine it for the final paper.
>
> **To address requested change 3**:
> - Comments the struggle of LLMs in geometric reasoning has been added in result analysis

---

### Decision · Action_Editor_LRHi · 2025-02-03

**Recommendation:** Reject

**Comment:**

While this paper introduces a novel application of LLM for floor plan generation using structured data and demonstrates promising results, all reviewers found the paper need significant improvements to meet the bar of an academically insightful paper. The overall informal and unclear, lacking the precision expected in academic papers.

**Audience:**

(1) The major audience who are interested in multimodal design, especial in floor plan generation. (2) The audience in LLM may found this work less interesting as insights to inspire better LLM training or inference can be added, such as the agentic approach or chain-of-thoughts can considered to address this hard problems, in addition to the current form of the work.

**Claims And Evidence:**

- This paper is interesting in its novel application in the domain of Floor Plan design using LLM. In particular, a language-based approach to this image-based design problem can inspire further research on different ways to improve the performance.

- The current form of this work lacks insights and depths, see the details below.

1. *The JSON-based data structure's effectiveness for LLM floor plan generation.*  This is a core claim about their technical approach. No experimental evidence was initially provided showing this structure works better than alternatives. Particularly problematic since it's presented as a main contribution. Only added comparison with natural language inputs after reviewer feedback. More advanced structure and prompt could be considered, eg, the inference techniques that scale well with test-time compute.

2. *Claim: The self-consistency metrics are effective measures of performance.* Reviewer x139 points out that scores being consistently near 1.0 suggests lack of sensitivity. No comparison of these metrics with existing models. Authors acknowledge they cannot compare numerical consistency with existing approaches.

3. *Performance on combined datasets (RPLAN and ProcTHOR)*. Authors only trained separately on each dataset. Combined training wasn't completed. Makes claims about cross-dataset performance without supporting evidence. Particularly notable since dataset handling is presented as an innovation.

4. *Claim: This is a "new dataset"*. The most significant contribution of this work to the machine learning community community is the novel dataset. However, Reviewer Fh9T points out this is inaccurate since they merely converted existing datasets. Authors don't directly address this overclaim in their response. The contribution appears to be more about data format conversion than creating new data.